health and disease and epidemiology, ecology

zoonotic disease, nutritional stress, bird, West Nile virus, agent-based models, host resistance

**Author for correspondence:**
J. C. Owen
e-mail: owenj@msu.edu

†Present address: Department of Biology, Emory University, Atlanta, GA, USA.

# Reservoir hosts experiencing food stress alter transmission dynamics for a zoonotic pathogen

J. C. Owen[1,2], H. R. Landwerlen[1], A. P. Dupuis II[4], A. V. Belsare[1,†],
D. B. Sharma[3], S. Wang[4], A. T. Ciota[4] and L. D. Kramer[4]

[1]Department of Fisheries and Wildlife, [2]Department of Large Animal Clinical Sciences, and [3]Center for Statistical Training and Consulting, Michigan State University, East Lansing, MI 48824, USA
[4]Griffin Laboratory, NYS Department of Health, Slingerlands, NY 12159, USA

JCO, 0000-0003-1383-4816; HRL, 0000-0002-3599-0580; APDII, 0000-0003-0703-9250;
AVB, 0000-0002-4651-0116; SW, 0000-0003-0931-7883; ATC, 0000-0001-6589-4728;
LDK, 0000-0002-8707-0778

Food limitation is a universal stressor for wildlife populations and is increasingly exacerbated by human activities. Anthropogenic environmental change can significantly alter the availability and quality of food resources for reservoir hosts and impact host–pathogen interactions in the wild. The state of the host's nutritional reserves at the time of infection is a key factor influencing infection outcomes by altering host resistance. Combining experimental and model-based approaches, we investigate how an environmental stressor affects host resistance to West Nile virus (WNV). Using American robins (*Turdus migratorius*), a species considered a superspreader of WNV, we tested the effect of acute food deprivation immediately prior to infection on host viraemia. Here, we show that robins food deprived for 48 h prior to infection, developed higher virus titres and were infectious longer than robins fed normally. To gain an understanding about the epidemiological significance of food-stressed hosts, we developed an agent-based model that simulates transmission dynamics of WNV between an avian host and the mosquito vector. When simulating a nutritionally stressed host population, the mosquito infection rate rose significantly, reaching levels that represent an epidemiological risk. An understanding of the infection disease dynamics in wild populations is critical to predict and mitigate zoonotic disease outbreaks.

## 1. Introduction

The emergence and transmission of zoonotic pathogens is increasing at an alarming rate and is expected to continue with globalization and anthropogenic land-use changes [1]. The risk of zoonotic pathogens to public health cannot be overstated, as evident from the pandemic from a novel coronavirus, 2019-nCoV, a zoonotic pathogen that originated in animals and spilled over into human populations. In fact, the most devastating disease outbreaks in humans are caused by zoonotic pathogens [1]. Our ability to predict the emergence and spillover of zoonotic pathogens into human populations and mitigate the impact of disease outbreaks relies on an understanding of the factors underlying infection and transmission dynamics in wildlife populations [2]. Land-use change can mediate the abundance and distribution of reservoir hosts and their capacity to maintain, move and transmit pathogens [3]. Hence, understanding the drivers of disease dynamics within the wild reservoir host populations is critically important to mitigating the health risk to wildlife, domestic animal and human populations [4].

Food limitation is a dominant source of stress for wild populations, and the availability of food is increasingly threated by human activities. Habitat alterations are almost always followed by changing availability and quality of food resources for wildlife which can alter host–parasite interactions and the

frequency and intensity of disease outbreaks. The mechanism(s) by which changing food resources may affect disease (transmission) dynamics through altering host–pathogen interactions include (i) behavioural changes that alter host's contact with other hosts or vectors [5,6], (ii) changes in host abundance and density through demographic shifts (increase and decrease in birth and death rates) and (iii) changing host physiology and immune defence [3,7,8]. Here, we examine the latter—the effect of food deprivation on host resistance to a viral pathogen.

A host's ability to maintain and engage their immune system depends on their ability to meet the energetic and nutrient requirements of immune responses [9], and a negative energy balance and/or malnutrition can significantly impair their immune function [10,11]. In particular, food deprivation may reduce cytokine production, cause immune organs to atrophy and reduce populations of immune cells [12]. In the absence of a properly functioning immune system, pathogen replication may be unchecked leading to higher pathogen loads and delayed recovery rates, and lengthening the time the host is infectious [8,13]. The duration and intensity (i.e. pathogen load) of this infectious state is one of the critical factors influencing transmission dynamics [14]; yet, despite the positive association between host nutrition and immune function [15], there are few studies on how the scarcity of food can affect a host's ability to defend itself against parasites [16,17].

The factors underlying variation in host infectiousness at the level of the population and individual are not well understood. The variation may be driven by the host's life history [18], host genetics, host–pathogen interactions and/or extrinsic factors [19,20]. In multi-host systems, interspecific variation in infectiousness has received the most attention due to its role in community-level disease dynamics by either diluting or amplifying pathogen transmission [21]. This interspecific variation in infectiousness is one determinant of a host's reservoir competency for arthropod-transmitted pathogen; another is the likelihood of the host being a source of an arthropod vector's blood meal and susceptibility to becoming infected [22]. More recently, attention has been focused on within-species variation in infectiousness [23–25], in which pathogen loads exhibit patterns consistent with the Pareto distribution with the top most infectious individuals being responsible for 80% of the total pathogen load [24]. If these highly infectious or 'supershedders' also make contact with susceptible hosts or vectors, they can alter disease dynamics and trigger superspreading events [26].

In this study, we use West Nile virus (WNV) and a common reservoir host, American Robin (*Turdus migratorius*; hereafter 'robin'), to advance our understanding of how an environmental stressor, food availability, affects within-species variation in host's infectiousness. In the 1990s, WNV became the most geographically widespread arbovirus globally with outbreaks occurring in Eastern Europe and Northern Africa, and its invasion and the subsequent human and bird outbreaks in the USA [27]. Human risk of WNV increases with enzootic transmission between birds and mosquitoes, which is driven by the reservoir competence of the amplifying hosts (i.e. birds). Robins are a ubiquitous and largely migratory landbird throughout much of North America. While they may not exhibit exceptionally high viral titres relative to other bird species [28–30], robins do develop moderately high viraemia and serve as a primary source of *Culex* mosquito blood meals, particularly in the eastern

USA [31–33]; and as such, they have been classified as 'superspreaders' of WNV [34,35].

Here we investigate an environmental determinant of host infectiousness—the availability of food. We tested the hypothesis that short-term food deprivation would make robins less resistant to WNV than birds not food stressed. Specifically, we predicted that robins deprived of food for a couple days prior to exposure to WNV would exhibit (i) higher viral titres, (ii) longer viraemia and (iii) higher morbidity and/or mortality than non-food-stressed individuals. We then use an agent-based model (ABM) approach to explore how changes in individual host infectiousness may impact population-level transmission dynamics of WNV.

Specifically, we developed an ABM to simulate enzootic transmission of WNV in an avian host (American robin)–mosquito vector (*Culex* spp.) system during the fall when migratory populations of robins are migrating south to their non-breeding grounds. The energetic cost of migration is high and migratory landbirds, including robins, rely on finding suitable stopover sites where they can rest and replenish depleted fat stores [36]. Yet, for myriad reasons, the availability of food can be unpredictable and/or insufficient [37,38], which not only has fitness consequences for the migrant [36], it can have implications for seasonal transmission dynamics of pathogens. Using empirical and experimental infection data, we use the ABM to test the effect of food stress on the population prevalence of WNV-infected *Culex* mosquitoes, termed the mosquito infection rate (MIR), which is a predictor of human risk [39]. Understanding the basis for this variation is needed to prevent disease outbreaks and minimize the global threat that emerging infectious diseases pose to human, animal and ecosystem health.

## 2. Methods

### (a) Study species and location

First-year (hatch year, HY) robins ($n = 42$) were captured during fall migration from 12 to 22 October 2018 using mist nets (36 mm mesh; 12 m × 2.6 m) in Laingsburg, MI (42.82, −84.38). Upon capture, we assessed bird's condition, body mass (±0.1 g), sex, and wing length, and presence of ectoparasites.

We transported birds to Michigan State University's Research Containment Facility's (URCF) and housed them in one of two identical rooms. Initially, we placed birds in individual wire cages (30 × 38 × 38 cm) until they acclimated, at which point we moved them to small aviaries (6″H × 2″W × 9″D) with two to four robins in each. Room temperature was maintained at 68–70ºF on average and the photoperiod that mimicked natural conditions of central Michigan (13 L : 11 D). Birds were provided *ad libitum* access to water throughout the entire experimental period. All the birds were fed a mixed diet appropriate for the species (see [23,40]). The nutrient composition of the diet was constant across the experiment. The birds were fed a modified version of the semisynthetic diet described by Johnston *et al.* [40] that all had the same macronutrient composition (49% carbohydrate : 30% protein : 17% fat). The amount of food was determined in previous pilot experiments as being sufficient for maintaining their condition/mass.

On 25 October 2019, robins were placed into bird holding boxes and transported via car from East Lansing, MI to Albany, NY, a 12 h trip. They were housed in one ABSL3 room at the New York State Health Department Wadsworth Center's Griffin Lab. Upon arrival, all birds were placed into individual wire cages identical to the ones described above and provided food

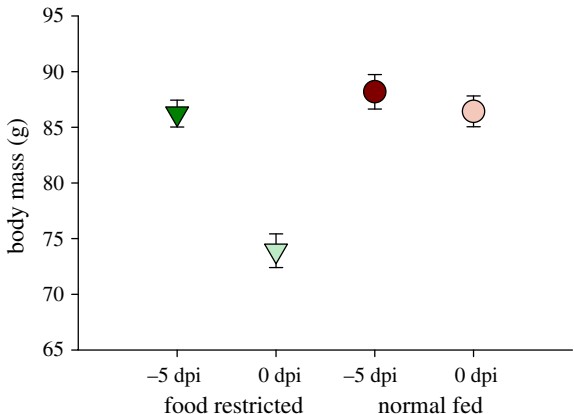

**Figure 1.** The experimental timeline. American robins were captured between 10 and 22 October and then transported to NY on 25 October, which, for purposes of describing the experiment, is denoted as −19 days post-inoculation (dpi). See 'Methods' for details. Created with Biorender.com. (Online version in colour.)

(as described above) and water, *ad libitum*. After one week in captivity in NY, we switched to feeding birds a fixed amount of food (30 g) to maintain their body mass (figure 1). The light schedule was identical to Michigan, and ambient room conditions were similar with average temperature and humidity of 70°F and 60%, respectively.

## (b) Experimental treatments

Prior to group assignment, we tested all robins for previous exposure to WNV by the plaque reduction neutralization test (PRNT; see below for methods). One robin tested positive for WNV antibodies (greater than 1 : 10), and two birds were inconclusive (titres of 1 : 10). Aside from these three individuals, we randomly assigned robins, stratified by sex and capture date to four treatment groups: two food restriction groups (i) 'FR_WNV' ($n = 10$; food deprived and infected with WNV) and (ii) 'FR_Sham' ($n = 11$; food restricted and inoculated with saline) and two normal-fed groups (iii) 'Norm_WNV' ($n = 10$; normal fed and WNV infected and (iv) 'Norm_Sham' ($n = 12$; normal fed and uninfected). The three robins that may have been previously exposed to WNV were randomly assigned to one of the two sham groups.

Forty-eight hours (−2 dpi; figure 1) prior to experimental infection, we stopped feeding the FR_WNV and FR_Sham groups and then resumed feeding birds on the day of inoculation (0 dpi). The Norm_WNV and Norm_Sham birds were fed normally, and all groups received water *ad libitum*. We weighed birds prior to and after food deprivation as well as throughout the entire experimental period (figure 1).

On 0 dpi (figure 1), birds in FR_WNV and Norm_WNV were inoculated subcutaneously in the cervical region with 0.1 ml of $10^5$ log PFU/ml of infectious WNV (strain WN02, 1986 in PBS diluent). The two sham groups were similarly inoculated with the PBS.

To assess viral titres, we collected whole blood (0.05 ml) daily through 6 dpi from the ulnar vein using a 25-gauge needle [41]. Blood was dispensed in BA-1 (M199 medium with Hank's salts, 1% bovine albumin, TRIS base (tris [hydroxymethyl] aminomethane), sodium bicarbonate, 2% fetal bovine serum and antibiotics) and stored at −80°C. Within a week, we quantified viral titres using the Vero cell plaque assay [42].

Additionally, to measure WNV antibodies, we collected whole blood (0.350 ml) prior to the experiment and on 14 dpi. The blood sample was stored at 4°C until antibody titres were assayed via PRNTs within two weeks of collection. Briefly, sera were diluted in BA-1 and heat inactivated at 56°C for 30 min. Sera were screened at a 1 : 10 dilution against WNV. Antibody titre was expressed as the inverse dilution of blood that neutralized 90% of the virus inoculum as measured by the virus-only control (no antibody; [43]) well. At 14 dpi, all the WNV-infected birds (control birds were held for a subsequent experiment not described here) were euthanized via $CO_2$ asphyxiation.

**Figure 2.** Average (±1 s.e.) change in American robin body mass between −5 and 0 dpi, following 48 h (−2 and −1 dpi) of food restriction for the FR groups. Food-restricted birds ($n = 21$) are denoted by green down triangles and normal-fed birds ($n = 21$) by red circles. (Online version in colour.)

## (c) Agent-based model

The ABM, AMRO_WNV_CULEX [44], was coded in the high-level language NetLogo [45]. AMRO_WNV_ CULEX advances using daily time steps and simulates the transmission of WNV between the American robin (avian host) and the mosquito vector (*Culex pipiens*; hereafter 'Culex'). Both these entities are modelled as individuals occurring in a hypothetical landscape, and interactions between the robin and *Culex* are simulated for 92 days (1 August to 31 October), when robins are actively migrating in mid-Michigan. The ABM approach facilitates the incorporation of heterogeneous and stochastic processes that influence WNV transmission in the real world, specifically robin migration events, host viraemia levels, mosquito population dynamics and *Culex*–robin interactions that underpin WNV transmission dynamics.

The purpose of the model is to provide a tool to run virtual experiments by implementing scenarios with varying levels of acute food stress experienced by robins during fall migration when there are large influxes of robins, particularly susceptible, HY birds and quantify the resultant proportion of WNV-infected *Culex* (or MIR) in the model landscape [39].

Here we provide the brief framework of the model, state variables, model parameters and model implementation. The ABM's full description following the ODD protocol [46] is shared in the electronic supplementary material.

### (i) Avian host (American robin)

In the model, robins have multiple states—age (HY; after-hatching year, AHY), infection status (susceptible, infected and

Proc. R. Soc. B **288**: 20210881

infectious (i.e. virus titres ≥4.0 log/pfu 0.1 ml)), infected and not infectious (virus titres less than 4.0 log/pfu 0.1 ml), or recovered with immunity, and stress condition (food stressed: Y or N). The robin abundance, migration phenology and age ratio are based on 10+ years of banding data near the location where robins were captured for the study. At the start, robins ($n = 500$) are randomly scattered in patches with an initial age ratio of 1 : 4 AHY : HY. The robin population increases on days 50 and 71 with the total robin population set at 1000 (2 : 3 AHY : HY) and 2500 (3 : 7 AHY : HY) birds, respectively.

Each week 95% of the robins, excluding individuals who are both food stressed and have viral titres exceeding 6 log pfu/ 0.1 ml (i.e. 'sick' robins), depart and are replaced by a new cohort of uninfected robins. If the 'sick' robins survive, they will depart the following week. Each susceptible robin has a probability of being bitten and progressing to an infectious state or non-infectious state as determined by the results of our experimental challenge in the current study. The per cent of infectious robins on 2, 3, 4 and 5 dpi is 90%, 70%, 50% and 30% for stressed robins and 70%, 30%, 10% and 10% for non-stressed robins, respectively. Mortality only occurs in 10% of the infectious, stressed birds on 5 dpi. The proportion of robins immune is age-dependent (values derived from the current study and WNV serosurveillance efforts in NY; AP Dupuis II, 2007 unpublished data).

### (ii) Mosquito

The simulation starts with 5000 mosquitoes (representing an initial 10 : 1 ratio of mosquitoes to birds). Mosquito mortality is simulated using the daily natural mortality probability derived from the literature [47]. The starting abundance of *Culex* is constant throughout the run; every *Culex* that dies, one adult uninfected female *Culex* is introduced into the population. Mosquitoes are susceptible, infected and non-infectious (extrinsic incubation period, EIP, 7–11 days) [48] or infected and infectious.

### (iii) Mosquito–host interaction

Here we assume a mosquito will take a blood meal from one (minimum)–five (maximum) robins with a bite rate of 0.17 per bird per day [47,49]. Mosquitoes will only become infected by biting a viraemic robin (transmission probability of 0.05–0.30) [50–52] and once infectious can only transmit virus (transmission probability of 0.8) [47] to robins after EIP.

### (d) Data analysis

All variables were tested for normality of distribution (Kolmogorov–Smirnov test) and the equality of variance (Bartlett's tests), and the $\alpha$ level was set for 0.05. When data were not normally distributed, we used the equivalent non-parametric test if available. Otherwise, we transformed the data to meet assumptions of normality, or when transformations did not normalize data, we report non-normality in our results below. To examine food and viral treatment effects on body mass and viral titres, we used two-way repeated-measures ANOVAs to examine between and within-group differences and their interactions using SigmaPlot (Systat Software, San Jose, CA).

Analyses and sample sizes varied with different comparisons. Prior to inoculation, we compared changes in body mass of the two groups according to food restriction treatment (food restriction, [FR], $n = 21$ and normal fed [Norm], $n = 21$). Post-inoculation, we compared viraemia response of the two WNV-inoculated groups (FR-WNV, $n = 10$ and Norm_WNV, $n = 11$). In each case, we used a repeated-measure, mixed model ANOVA to test for differences between and within groups. For significant main effects, we used the Bonferroni-corrected pairwise multiple comparison procedure. We calculated an infectious index for each bird by calculating the area under the curve for their viraemia profile above the 4 log pfu/0.1 ml (or commonly

reported as 5 log pfu/ml) blood threshold. While this threshold of infectiousness varies with different mosquitoes [50,52,53], viral titres of 4 log pfu/0.1 ml and above are likely to infect blood-feeding *C. pipiens*, the primary vector in the northeastern and midwestern USA [54]. Differences in the AUC according to food treatment were analysed using a one-way ANOVA.

Using the ABM, we tested the effect of changing proportion (0.0, 0.10, 0.25 and 0.50) of stressed robins in the population on MIR from 1 August to 31 October. Each scenario yielded the number of WNV-infected *Culex* per 1000 mosquitoes (i.e. MIR; [39]) per time step based on 100 replicate simulations for each scenario. We analysed the data in two distinct phases that address two different aspects of WNV transmission dynamics: early in the migration season (referred to as Phase 1; 1 August–19 September) when the virus is locally amplifying in the simulation and Phase 2 (20 September–31 October) when there are large influxes of robins during migration.

The analysis approach for the model output was determined through visualization of the data by day and week for each phase, which revealed linear trend in Phase 1 and quadratic trend in Phase 2 (see the electronic supplementary material). Using the AIC model selection criterion, we determined the best model, which we report in this study. All ABM analysis was conducted using the R statistical software [55,56], with the glmmTMB [56] package used to account for heteroscedasticity.

## 3. Results

All the WNV-inoculated birds were successfully infected as evident by viraemia and production of WNV-specific antibodies by 14 dpi. None of the birds died during the experimental infection period; however, two of the birds in the FR_WNV group exhibited clinical signs of WNV—including lethargy and anorexia (lack of appetite). These overt symptoms became notable on 3 dpi and disappeared by 6 dpi, with both birds making a full recovery as evident by normal behaviour and mass gain.

### (a) Food treatment and mass changes

We examined the effect of food restriction on change in body mass (between −5 and 0 dpi; figure 1), prior to virus inoculation for both FR ($n = 21$) and Norm ($n = 21$) groups. There was a significant main effect between groups (FR and Norm; $F_{1,82} = 13.92$, $p < 0.001$; figure 2) and across time within groups ($F_{1,82} = 181.23$, $p < 0.001$) with a statistically significant interaction ($F_{1,82} = 106.24$, $P < 0.001$). Body mass on −5 dpi did not differ between groups ($p = 0.339$), but it did differ post-food restriction (0 dpi; $p < 0.001$). Within groups, body mass significantly dropped between −5 and 0 dpi for both groups (Tukey's test both $p < 0.001$); however, the mean drop in mass was 1.76 g (±2.62 s.d.) for Norm and 12.32 g (±3.81 s.d.) for FR birds. When we examined mass changes for the four treatment groups for −5, 0, 2 and 4 dpi, there was a significant main effect of group ($F_{3,152} = 18.61$, $p < 0.001$) and date ($F_{3,152} = 9.049$, $p < 0.001$) and no significant interaction effect. *Post hoc* analyses reveal no changes in body mass over time for the non-food-restricted groups (Norm_CTRL and Norm_WNV). Food-restricted birds all regained mass post-food restriction, regardless of infection status (see the electronic supplementary material for additional results on body mass).

### (b) Food treatment and viral infection

Birds that were food deprived for the 48 h preceding infection (FR_WNV) had higher and prolonged viraemia compared

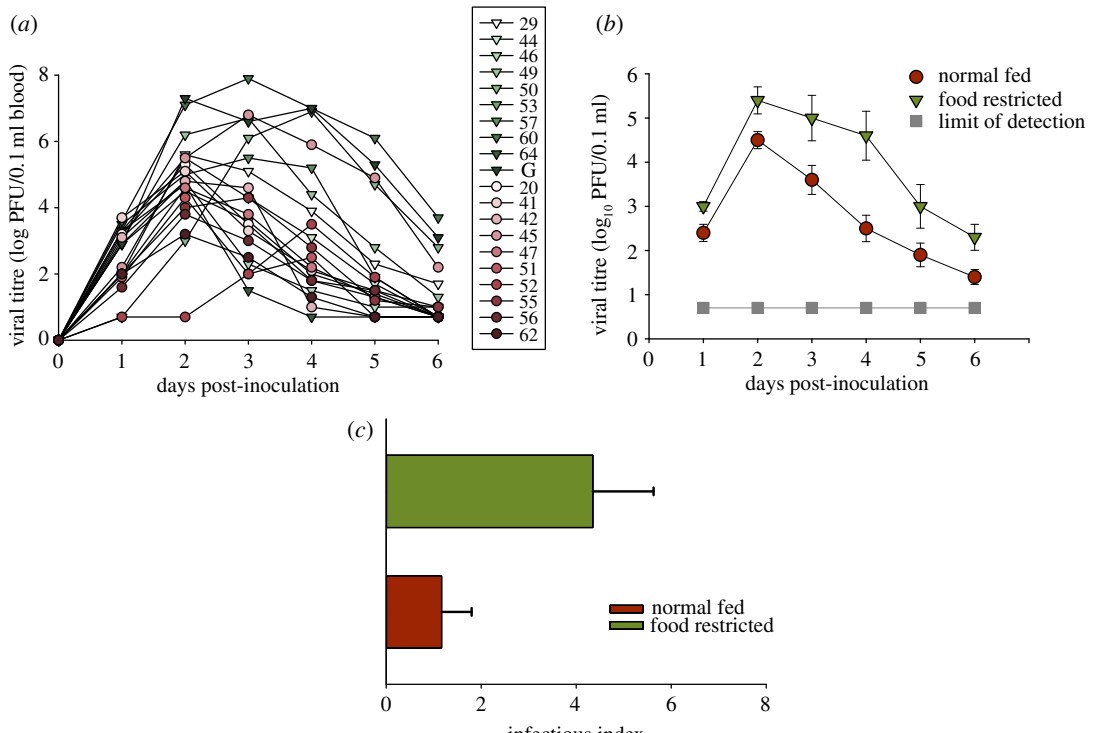

**Figure 3.** (*a*) WNV titres (log PFU/0.1 ml blood) for experimentally infected American robins fed normally (red circles; $n = 11$) and robins food-deprived for 48 h prior to inoculation (green down triangles; $n = 10$). (*b*) Average ($\pm 1$ s.e.) virus titre per log PFU/0.1 ml blood for normal (red circles) and food-restricted (green down triangles) robins. Virus titres below 0.7 log pfu/0.1 ml are undetectable (grey squares) via Vero cell plaque assay. (*c*) Mean ($\pm 1$ s.d.) infectious (i.e. capable of infecting a biting mosquito) index for the two groups. (Online version in colour.)

with Norm_WNV birds. All the FR_WNV (10/10) became infectious (i.e. 4.0 log/0.1 ml) for at least 1 day compared with only seven of the 10 Norm_WNV birds (figure 3*a*). Additionally, FR_WNV had consistently higher viral titres (figure 3*b*; $H_{1,108} = 20.66$, $p < 0.001$), were infectious longer (200% more infectious days, 24 days for FR_WNV and 12 days for Norm_WNV; $t = -2.48$, d.f. = 18, $p = 0.023$) and had a higher infectious index (i.e. AUC) than the Norm_WNV group (figure 3*c*; $U = 18.0$, $N = 10$, $p = 0.017$).

All WNV-infected birds, regardless of the treatment group, seroconverted by 14 dpi, with titres of WNV-specific neutralizing antibody (PRNT$_{90}$) ranging from 80 to 5120. While there was no significant difference in antibody titre by food restriction treatment (ANOVA on ranks; $H = 3.171$, $p = 0.075$), there was a positive association between virus titre and PRNT$_{90}$ titres ($R^2 = 0.54$, $N = 20$, $p < 0.001$), a relationship driven mainly by two FR birds with the highest viral titres.

## (c) Agent-based modelling results

Changing the proportion of the American robin population experiencing food stress significantly affected the prevalence of infected *Culex* spp. (figure 4). From the models compared using the AIC in Phase 1, the best model was a model of weekly aggregated data considering unequal variances and a spline function over time. The results of this model show that the food stress level ($F_{1,11188} = 5601.37$, $p < 0.001$), week ($F_{1,11188} = 46.86$, $p < 0.001$) and their interaction ($F_{3,11188} = 221.10$, $p < 0.001$) were all significant, indicating that MIR varied with time and food stress level. Further pair-wise comparisons with Bonferroni correction show that the four food stress levels have a significant effect on MIR (electronic supplementary material, table S2).

From the models compared using the AIC in Phase 2, the best model was a model of daily data that included a linear term of food stress levels as a main predictor, linear term of day as a covariate, quadratic term of day and their interactions. The model found significant results of all terms in the model (day, $F_{1,11188} = 116.81$, $p < 0.001$; food stress, $F_{3,11188} = 5827.85$, $p < 0.001$; Day$^2$, $F_{1,11188} = 121.55$, $p < 0.001$; day × food stress, $F_{3,11188} = 13.23$, $p < 0.001$; food stress × day$^2$, $F_{3,11188} = 10.77$, $p < 0.001$). Pair-wise comparisons with Bonferroni correction show that all four stress levels are significantly different with each other (electronic supplementary material, table S2).

## 4. Discussion

Food restriction, even when for short duration and prior to infection, has a significant effect on the duration and magnitude of infectiousness of WNV-infected American robins. When deprived of food for 48 h, robins developed higher viral titres for longer duration than normal-fed birds. Moreover, the increase was epidemiologically significant with the total number of infectious days in food-deprived birds being 200% greater than the normal group. Its significance is further illustrated by the ABM, showing the positive association between food-stressed robins and the infection rate of mosquitoes. Even when only 10% of the population was 'food stressed', the infection rates for mosquitoes significantly increased. Furthermore, the MIR in the simulation reached levels associated with previous WNV epidemics [57]. American robins have already been considered one of the superspreaders of WNV [32,34,58,59], and here we show that limiting food resources during the migratory period further

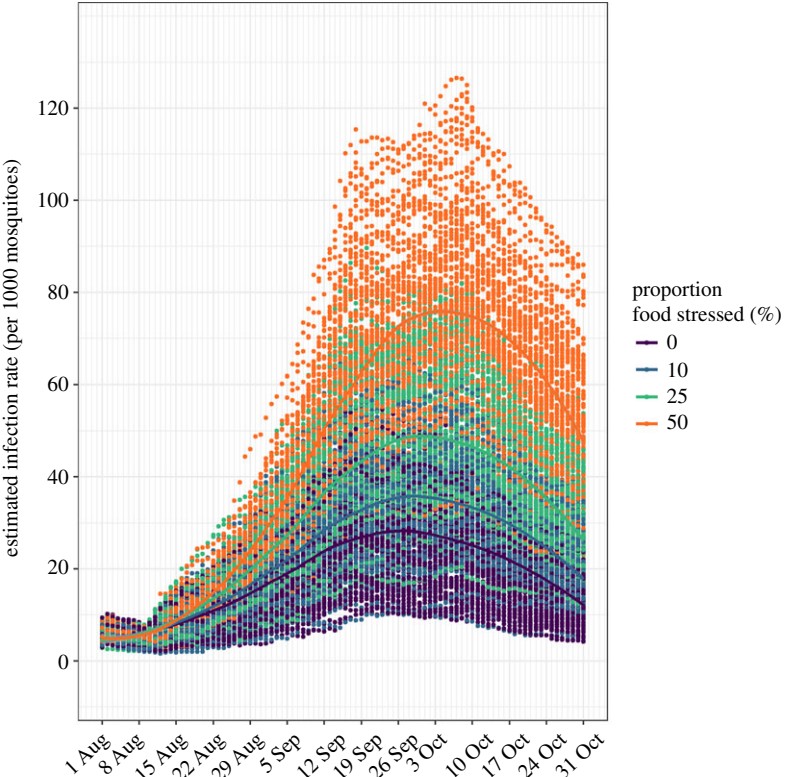

**Figure 4.** Daily estimated infection rate (no. of infected mosquitoes per 1000 mosquitoes) for four different scenarios with 0, 10, 25 and 50% of the American robin population experiencing food stress (100 simulations per scenario). (Online version in colour.)

magnifies their role in the amplification of WNV transmission. Our experimental results support theoretical models that suggest food scarcity may lead to superspreading events due to within-host effects [60].

The availability of food is a universal stressor and, regardless of the factors driving that availability, it is an important limiting factor acting on wildlife populations. Nonetheless, while it is well understood that food deprivation and malnutrition can alter host defences to pathogens [61–63], few studies have investigated the effect of an acute lack of food prior to exposure to a pathogen, particularly in a wildlife reservoir. The state of the host's nutritional reserves at the time of infection is a key factor influencing viral infection, as is the duration and severity of the food restriction [64]. Our findings are consistent with other studies that show a link between short-term food deprivation and reduced immune function [16,65]. In laboratory mice, food deprivation, prior to and during the early part of the infection had large effects on host resistance to a fungal pathogen [16]. Moreover, in that same study, they found that the biggest effect on host resistance was when deprivation began prior to infection and continued through the first-day post-infection [16].

The relationship between food stress, immunity and resistance to pathogens is inherently complex and can vary with pathogen or host-related factors such as mode of transmission and/or site of binding and replication or the nature of the food stress (e.g. duration, nutrient or caloric modifications and restriction/supplementation). In mallards (*Anas platyrynchous*) infected with low-pathogenic avian influenza virus (LPAIV) chronic (30 days) food restriction affected viral shedding but, contrary to the current study, pathogen load progressively decreased with the severity of food limitation [17]. In swine influenza-infected mice, mice exhibited a cyclic susceptibility to viral infection relative to the duration

of protein restriction, with mice on low protein diets for two and eight weeks showing the highest susceptibility, while mice restricted for moderate duration (four and six weeks) showing the lowest viral titres [64]. This oscillation of viral titres may be associated with different stages of gluconeogenesis [64]. In the current study, we just altered access to food; however, the lack of some nutrients, such as protein, which is an important modulator of host immunity, can have a greater effect on host resistance to pathogens (e.g. [62,66,67]). Furthermore, where a pathogen invades and colonizes the host tissues can affect the relationship between host resistance and the quantity and quality of food. In the two aforementioned studies [17,64], the influenza virus binds and replicates within the intestinal tract. Dietary changes can strongly influence the physiological and metabolic processes of tissues in the gastrointestinal tract [68], which in turn could affect the availability of tissues expressing receptors for the pathogen.

Too little food is just one form of poor nutrition. Overnutrition or obesity may cause delayed or suppressed antiviral responses, as seen in humans in which obesity as a pre-existing condition is shown to lead to worse disease outcomes [69]. While obesity is rarely a problem for free-living wild populations of birds, there are examples of augmented food resources leading to higher infection rates. For instance, food provisioning, such as supplemental feeding with bird feeders, is associated with higher infection rates, an outcome most likely a consequence of increased in contact rates as individuals congregate at feeding stations [7,70,71]. However, once infected, the negative effect of food provisioning may be mitigated by the positive effects of food resources on host immunity [72], thereby suppressing pathogen replication [7,70].

American robins are considered one of the most important avian reservoirs for WNV [31,35,58] and have been the focus

of several previous experimental infection studies [29,73]. The viraemia profiles of the normal-fed robins in our study are similar to what has been observed previously ([73,74]; unpublished data by JC Owen & AP Dupuis II 2019). Furthermore, these experimental infection studies document low WNV-associated mortality, with birds only dying when their virus titres exceed 7.5 log pfu/0.1 ml [73]. While none of the robins succumbed to the virus in this study, some food-deprived robins did exhibit significant morbidity. Two FR_WNV robins (60 and G), with the highest titres in the whole study, did exhibit lethargy, anorexia, and were 'puffed' up, from 3–6 dpi, all common signs of diseased birds. While they ultimately recovered and survived the infection, their fate in the wild would be less certain. In the Norm_WNV, one bird (45) did have high viral titres similar 'G' and '60' but never exhibited any overt signs of disease. Similarly, in a previous study with WNV-infected northern cardinals (*Cardinalis cardinalis*), corticosterone-implanted birds had higher morbidity and mortality, but similar viraemia profiles as the empty-implanted (control) birds [23].

A limitation of challenge experiments is knowing whether the treatment effects observed in captive experiments translate into epidemiologically important outcomes in nature. ABMs are being increasingly used to investigate complex host–pathogen systems to better understand the spread and persistence of pathogens in host populations and are particularly useful for elucidating complex causal effects. To illustrate the significance of food deprivation on disease dynamics of WNV, we developed an ABM to explore how changing individual-level infection dynamics in avian hosts affects infection rates of *Culex* mosquito (vector), which is a predictor of human infection risk. The model demonstrates that increasing the proportion of the robin population experiencing 'food stress' leads to an increase in the prevalence of infected *Culex* mosquitoes, with average weekly infection rates from six to 78 infected individuals per 1000 mosquitoes. Furthermore, with the weekly turnover in robin population during migration coupled with the larger movements of robins in mid to late September and early October, the transmission cycle stays elevated and peaks when robin numbers are highest (figure 4). The migration phenology depicted in this model characterizes the timing of robin movements in mid-Michigan and may not be applicable across its entire range. However, the relationship between food stress in the wildlife reservoir and the infection rate in the mosquito vector may reflect the patterns observed in other regions and even other host–vector–parasite interactions.

While it is difficult to link a specific population infection rate of mosquitoes 'entomological risk' [75] to a level of risk to humans, there is evidence that higher infection rates in mosquitoes correspond positively with the number of human cases of WNV [39,76]. The infection rate of *Culex* in the simulation, while offset temporally, does reflect the range observed in nature [77–79]. Here we are illustrating the relative increase and not interpreting the absolute value of the MIR. Entomological risk is a function of mosquito population infection rate, abundance and host-feeding preferences [34]. The contact rate between mosquitoes and avian hosts is not random; blood-seeking females will preferentially feed on some species more than others [31] and this variation in vector–host-feeding preferences is a key factor driving transmission dynamics of WNV [34]. In the northeast and midwestern USA, American robins are consistently the most common source of avian-derived blood meals for *C. pipiens*, regardless of their relative abundance within the community [31,33,35,58]. Given these host-feeding preferences for robins, our bird-to-mosquito contact rate is likely conservative.

Here we ask whether having food-stressed avian hosts in a community affects WNV transmission. To gain insights, we created a model depicting a simple system with WNV transmission occurring in a community with only two species—the American robin and a *Culex* mosquito. In nature, WNV transmission occurs within the context of a multi-host–multi-vector community, and within that community are a myriad of bird and mosquito species that vary in their reservoir and vector competence, respectively [57]. A suite of other factors can influence host–vector–pathogen interactions. For instance, host behaviour can influence vector–host contact rates, with highly viraemic birds becoming lethargic, as shown in our experiments with food-stressed robins, and exhibit fewer anti-mosquito defensive behaviours allowing mosquitoes to successfully complete their blood meal [80]. The empirical data and theoretical model collected and developed in this study further support the role of robins as WNV superspreaders, which is further amplified when migrating robins experience an environmental stressor, such as limited and unpredictable food availability.

Humans have dramatically altered the environment through technological advances and a population that has more than doubled in the last 50 years, moving us into a new geological period, the Anthropocene. The emergence and spread of zoonotic diseases are just one of the many unintended consequences of globalization and land-use change. Using both empirical and theoretical approaches, we demonstrate the effect of resource availability on within-host–parasite processes and how a change in host infectivity can alter community transmission dynamics and risk to human health.

**Ethics.** All procedures and methods were approved by IACUC at Michigan State University (AUF 12-16-211-0) and Wadsworth (18-412) and Federal and State Scientific Collection Permits to J.C.O. (SC1386, MB194270) and Master Banding Permit (no. 23269).

**Data accessibility.** Data are available from the Dryad Digital Repository. American robin West Nile viral titre data and R script and for agent-based modelling data: https://doi.org/10.5061/dryad.ksn02v74s [81]. American robin banding data: https://doi.org/10.5061/dryad. 3xsj3txgq [82].

**Authors' contributions.** J.C.O.: conceptualization, data curation, formal analysis, funding acquisition, investigation, methodology, project administration, resources, supervision, validation, writing-original draft, writing-review & editing; H.R.L.: data curation, investigation, methodology, writing-review & editing; D.B.S.: formal analysis, methodology, visualization, writing-original draft, writing-review & editing; S.W.: formal analysis, methodology, visualization, writing-original draft, writing-review & editing; A.T.C.: methodology, resources, supervision, writing-review & editing; L.D.K.: resources, supervision; A.P.D.: data curation, investigation, methodology, writing-review & editing; A.V.B.: conceptualization, data curation, formal analysis, methodology, writing-review & editing.

All authors gave final approval for publication and agreed to be held accountable for the work performed therein.

**Competing interests.** We declare we have no competing interests.

**Funding.** The work was supported by NSF IOS-1350772 to J.C.O., MSU AgBioResearch and NYS Department of Health.

**Acknowledgements.** We thank Z. Spodek, A. Cole, A. Shoffner and D. Owen who all helped make this research possible.

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
