## [Peer Review File · Proceedings of the Royal Society B: Biological Sciences]

Review History

Decision letter (RSPB-2021-0642.R0)

31-Mar-2021

Dear Dr Owen:

I am writing to inform you that your manuscript RSPB-2021-0642 entitled "Reservoir hosts experiencing food stress alter transmission dynamics for a zoonotic pathogen." has, in its current form, been rejected for publication in Proceedings B.

However, the editor will reconsider your paper if you address the comments provided.

Sincerely,

Dr The Proceedings B Team

Board Member:

Comments to Author(s):

This paper investigates the effects of resource restriction for pathogen resistance and consequences for disease dynamics focusing on West Nile virus (WNV) infection in American Robins. Robins are an important reservoir host for WNV in North America and this paper uses a food deprivation experiment to quantify the effect of a reduction in resources on WNV viremia and duration of infectiousness. Next, based on these the experimental results the authors develop an agent based model that tracks transmission between migrating robins and mosquitoes to examine how nutritional stress affects WNV transmission dynamics of WNV. Model results suggest that nutritional stress of hosts can significantly increase mosquito infection rates, a result which has potential repercussions for human infection risk. This study nicely combines experimental and modeling approaches to address an interesting question about how physiological changes within hosts, driven by changes in the environment, can influence epidemiological outcomes. As such, I think this paper would be of broad interest to readers of the journal. However, the paper currently lacks sufficient detail about the modeling approach within the main text. Most of the relevant modeling required to understand what was done appear in the supplement, and this detracts significantly from the paper. To be reviewed, the methods of the paper would need to be sufficiently described in the main text.

Author's Response to Decision Letter for (RSPB-2021-0642.R0)

See Appendix A.

RSPB-2021-0881.R0

Review form: Reviewer 1

Recommendation

Accept with minor revision (please list in comments)

Scientific importance: Is the manuscript an original and important contribution to its field?

Excellent

General interest: Is the paper of sufficient general interest?

Excellent

Quality of the paper: Is the overall quality of the paper suitable?

Good

Is the length of the paper justified?

Yes

Should the paper be seen by a specialist statistical reviewer?

No

Do you have any concerns about statistical analyses in this paper? If so, please specify them explicitly in your report.

No

It is a condition of publication that authors make their supporting data, code and materials available - either as supplementary material or hosted in an external repository. Please rate, if applicable, the supporting data on the following criteria.

Is it accessible?

Yes

Is it clear?

Yes

Is it adequate?

Yes

Do you have any ethical concerns with this paper?

No

Comments to the Author

I found this paper to be interesting and exciting, and worthy of publication in Proceedings B. My comments below are focused on the reporting of the agent-based model in the main text and results. Overall this is well described, but I have a few suggestions and clarifying comments below, with the most important point to address relating to assumptions about the transmission rate and how mosquito abundance changes with bird abundance.

1. Clarify assumptions about mosquito abundance. In the supplement (c-mortality) you say that the abundance of mosquitoes is fixed over time, but in the main text and elsewhere it says that the relative abundance of mosquitoes to birds is fixed at a 10:1 ratio. So what happens to mosquito abundance during the large influx of robins on day 50 and day 71? If you assume that mosquito abundance also jumps with the influx of birds on these days you should make this assumption explicit, and justify why it's reasonable to assume that mosquito abundance tracks bird abundance through time in this way. You should briefly discuss the potential consequences of this assumption for transmission in the discussion (see point 3 on transmission rate below)
2. For replicability, it's important to show the functional forms used to describe any transition probabilities relating to demography and infection that are not constant per capita rates in the supplement (e.g. how exactly are the effects of density-dependent mosquito mortality and temperature-dependence in transmission modelled?).
3. Crucially, the functional form of the transmission rate between host and vectors needs to be specified. Did you use the standard Ross-McDonald transmission function for vector-borne transmission, which assumes that bites are distributed among hosts and thus the number of bites per host decreases with increasing bird density? Or alternatively did you use a simple mass-action term (and if so, justify why)? This specification is important because it relates directly to what happens to transmission following an influx of hosts. If an influx of new birds arrives and the number of mosquitoes doesn't change (see point 1) transmission could actually decrease because the number of bites per bird decreases.
4. In the description of the bird hosts in the main text and supplement, specify the infection status of all immigrating birds – is it assumed that all immigrants are uninfected?
5. Clarify the statement about 'herd immunity' in p6 of the parameterization section of the supplement: is the proportion of immune birds updated in the model each week applied only to new arrivals? I think in an earlier section you say that birds recover after 6 days. Also, because 'herd immunity' typically refers to the population-level phenomenon arising from having immune individuals rather than the % immune per se, I would rephrase to describe as 'the proportion of birds in the recovered and immune class'.
6. The model assumes that mosquito biting preferences, and mosquito mortality, are not influenced by the infection or nutritional status of the host. This could be worth specifying given empirical work demonstrating that these assumptions don't always hold, and thus could influence prevalence in mosquitoes

Minor comments

Main text

Line 169: shorten the title of this section to 'agent based model'

Line 172: add 'the' before American robin (and check for other instances of missing definite articles in the modeling and results section and supplement)

Line 240: define MIR at its first use

Lines 242-246: Modify to use a colon after the first sentence and a semicolon between the descriptions of phases 1 and 2.

Lines 306 and 314: I found the sentences about the stress levels difficult to interpret - could this be rephrased?

Supplement

page 6. I didn't understand the very last sentence on this page "With a 1-host."

Review form: Reviewer 2

Recommendation

Accept with minor revision (please list in comments)

Scientific importance: Is the manuscript an original and important contribution to its field?

Excellent

General interest: Is the paper of sufficient general interest?

Excellent

Quality of the paper: Is the overall quality of the paper suitable?

Good

Is the length of the paper justified?

Yes

Should the paper be seen by a specialist statistical reviewer?

No

Do you have any concerns about statistical analyses in this paper? If so, please specify them explicitly in your report.

No

It is a condition of publication that authors make their supporting data, code and materials available - either as supplementary material or hosted in an external repository. Please rate, if applicable, the supporting data on the following criteria.

Is it accessible?

No

Is it clear?

No

Is it adequate?

No

Do you have any ethical concerns with this paper?

No

Comments to the Author

I enjoyed reading this study by Owen et al. that examined how short-term food restriction influences West Nile virus infection dynamics for American robin reservoirs. This is a critical question because variation in food availability is arguably one of the most universal ecological stressors in natural populations, but it's a challenging stressor to study outside of experimental manipulations such as the one done here. Thus, we know very little about how variation in food access, including short-term food restriction, influence infection dynamics in wildlife. In this case, understanding infection dynamics in wildlife has important consequences for human health, given that the virus studied here is zoonotic in nature. Overall this was a very well done experiment and I particularly liked the integration of the experimental data with a mathematical model to extrapolate the within-host effects detected to disease dynamics more broadly. I have some comments that I hope will improve clarity and impact of the manuscript.

Broader comments

The authors place the study in the context of anthropogenic environmental change in several places throughout the manuscript. While I appreciate that global change will likely exacerbate variation in resource availability, what I think is particularly important about resource availability (that doesn't really come through in the manuscript, in my opinion) is that is a relatively universal stressor that can occur even in the absence of anthropogenic effects. I think the manuscript would be even stronger if that aspect came through. I would also like for the connection between food availability and migration to be made stronger in the manuscript- the authors focus on robins during the fall for good reason, but don't really build the conceptual links between their study design and when robins might experience food restriction "naturally". For example, at the end of the introduction you nicely set up your experiment and model but there is no direct connection made between robin ecology (what the likely periods of food restriction in the wild and how those map onto demography, migration) and WNV spread. Some of this comes later when you set up the model (Lines 182-185) but I think it would be much stronger to set this all up in the intro since it is really the motivation for the study design. You also mention later on that 20 Sept- 31 October is the local viral amplification phase but I don't think this comes up until Line 244-245. It would be great to have this upfront so the motivation for the timing of your experiment is understood.

I really liked that you based your model on empirical data, including banding data and serosurveillance data from nearby. I wasn't sure if some of that data needs to be analyzed or presented descriptively more formally (in the supplement), given that it motivates the model, or whether that is done elsewhere and thus not needed here?

Minor comments:

Line 16: I think "as" should be "at" in this case.

Line 27-28: duplication of "and predict"

Line 42: I wasn't sure what "top level" meant in this context- is this akin to "top down"?

Line 54-55: This is minor but this is the first use of "host resistance"- it could be clarifying to explicitly link this term back to mechanism 3 where you don't explicitly use the term resistance.

Line 57: might be helpful to add "of immune responses" after "requirements"

Line 63: this is the first use of "infectiousness"- might be helpful somewhere to clarify that here you use viral loads as a proxy for infectiousness.

Line 66: this was a bit of a rough transition for me when reading... could you instead link better to prior paragraph by starting with something like "The variation in infectiousness caused by nutrition is key to understand because variation in infectiousness at both the level of the population and individual is not well understood."

Line 77: I suggest adding "or vectors" after "susceptible hosts"

Line 81: might be helpful to clarify that here that you are looking within-species by saying "among-individual variation in..." or "within-species variation in host infectiousness" instead of just "host's infectiousness".

Line 84: I suggest deleting the word "amplified" here.

Line 90: I wouldn't have a dash between "environmental" and "determinant"

Very end of Line 140: Given that 3 birds were treated slightly differently (for good reason), I suggest adding "Aside from these three individuals,..." just before "we randomly assigned robins..."

Lines 147-151: It's clear in the figure but it would be good to make it clear in the text that birds were re-fed again starting at inoculation or just before. It is unclear in the text itself here.

Lines 200-201: I wasn't clear what happens to the robins that are both food stressed and have high viral titres- do they die? This makes sense based on your results but it isn't clear in text whether they are just not departing for migration or are dying.

Line 206: I suggest adding "that are" in front of "immune"

Line 211: I suggest adding "are" in front of "replaced"

Line 225: Technically I think this would be a mixed linear model, since a traditional ANOVA doesn't allow for random effects. But maybe this is incorrect and I just haven't heard of whatever model you used... was this done in R?

Line 252: you mention "multiple models" - is the model set listed somewhere? and how was it determined?

Line 268-269: can you clarify what you mean by "between" and "within" groups? Make clear what the "group" refers to here.

Line 272: if you give the mean in the text, I would also give some kind of metric of uncertainty like standard deviation.

Lines 277-283: there are a lot of different mass analyses- I see that they slightly differ and perhaps are all important to include but I wonder if it could be simplified in the text and some of the analyses moved to supplement so that it doesn't distract from the key analyses of how food restriction affected infection responses.

Line 290: "infectious longer" - is the unit in days? I don't think I saw how this was quantified- just the number of days over the infectious index of 4?

Line 300-315: I would add "food" in front of stress throughout this entire section just to clarify what type of "stress" you mean. Stress is such a vague term and you imposed a very specific form of it.

Lines 307-308: awkward wording of this sentence.

Line 315: also a bit awkward- perhaps different "from" each other rather than "with."

Line 322: it wasn't clear to me where the 200% came from in the results section. I like doing percentages for easier interpretation but it would help if it were set up in the results section to see what result generated this.

Line 326: I liked how you made the link to levels associated with previous WNV epidemics. Could that level be somehow added to the graph to make it visually obvious?

Line 334: I think should be "at" instead of "as":

Line 337: "biggest effect" relative to what? perhaps just say large effects to be safe.

Line 377: could you add a "were" before "puffed up"

Line 380: seems like a few words are missing (similar TO, maybe?)

Lines 383: this is an interesting link with the cort implants. Do you think cort may mediate some of the effects seen here? Do cort levels go up with food restriction? You may not want to speculate on this but the parallels were quite interesting.

Line 385: "translates" should be "translate" here

Line 391: Awkward wording- maybe say "demonstrates that increasing the proportion of the robin population experience..."

Line 402: remove either "of" or "in" at end of line

Figure comments

Figure 2 caption. Can you put the dates in for the calculation of proportional change in body mass (was that day 19 minus 14, for example)?

Figure 2. Could the treatments be organized to put the food restricted treatment next to each other, and/or with the same color?

Figure 3. I like the idea of showing the data for individual birds to see variability (Panel A) but it is hard to see which birds are in which treatment- can the points themselves be made slightly larger? Also the fact that the triangles and circles represent the treatment assignments in Panel A is not clear anywhere (needs to be in legend) and differs from the use of triangles in Panel B. To avoid confusion I would use a different type of symbol for the limit of detection and keep the triangles representing food restriction in Panel B as in Panel A.

Figure 3. Can you clarify the Y-axis label that it is per 1000 mosquitoes? Also, can Proportion stressed say Proportion "Food" Stressed for clarity?

Decision letter (RSPB-2021-0881.R0)

26-May-2021

Dear Dr Owen:

Your manuscript has now been peer reviewed and the reviews have been assessed by an Associate Editor. The reviewers' comments (not including confidential comments to the Editor) and the comments from the Associate Editor are included at the end of this email for your reference. As you will see, the reviewers and the Editors have raised some concerns with your manuscript and we would like to invite you to revise your manuscript to address them.

Research ethics:

Use of animals and field studies:

It is a condition of publication that you make available the data and research materials supporting the results in the article (<https://royalsociety.org/journals/authors/author-guidelines/#data>). Datasets should be deposited in an appropriate publicly available repository and details of the associated accession number, link or DOI to the datasets must be included in the Data Accessibility section of the article (<https://royalsociety.org/journals/ethics-policies/data-sharing-mining/>). Reference(s) to datasets should also be included in the reference list of the article with DOIs (where available).

If you wish to submit your data to Dryad (<http://datadryad.org/>) and have not already done so you can submit your data via this link [http://datadryad.org/submit?journalID=RSPB&manu=\(Document not available\)](http://datadryad.org/submit?journalID=RSPB&manu=(Document%20not%20available)), which will take you to your unique entry in the Dryad repository.

Please submit a copy of your revised paper within three weeks. If we do not hear from you within this time your manuscript will be rejected. If you are unable to meet this deadline please let us know as soon as possible, as we may be able to grant a short extension.

Best wishes,
Professor Hans Heesterbeek
<mailto:proceedingsb@royalsociety.org>

Associate Editor Board Member

Comments to Author:

This manuscript has now been evaluated by two expert reviewers. Both reviewers agree that this is an exciting study, however, they each identify components of the manuscript that require clarification, including some of the assumptions of the model and the overall framing of the study. The reviewers also provide a number of specific suggestions that the authors should consider.

Reviewer(s)' Comments to Author:

Referee: 1

Comments to the Author(s).

I found this paper to be interesting and exciting, and worthy of publication in Proceedings B. My comments below are focused on the reporting of the agent-based model in the main text and results. Overall this is well described, but I have a few suggestions and clarifying comments below, with the most important point to address relating to assumptions about the transmission rate and how mosquito abundance changes with bird abundance.

1. Clarify assumptions about mosquito abundance. In the supplement (c-mortality) you say that the abundance of mosquitoes is fixed over time, but in the main text and elsewhere it says that the relative abundance of mosquitoes to birds is fixed at a 10:1 ratio. So what happens to mosquito abundance during the large influx of robins on day 50 and day 71? If you assume that mosquito abundance also jumps with the influx of birds on these days you should make this assumption explicit, and justify why it's reasonable to assume that mosquito abundance tracks bird abundance through time in this way. You should briefly discuss the potential consequences of this assumption for transmission in the discussion (see point 3 on transmission rate below)
2. For replicability, it's important to show the functional forms used to describe any transition probabilities relating to demography and infection that are not constant per capita rates in the supplement (e.g. how exactly are the effects of density-dependent mosquito mortality and temperature-dependence in transmission modelled?).
3. Crucially, the functional form of the transmission rate between host and vectors needs to be specified. Did you use the standard Ross-McDonald transmission function for vector-borne transmission, which assumes that bites are distributed among hosts and thus the number of bites per host decreases with increasing bird density? Or alternatively did you use a simple mass-action term (and if so, justify why)? This specification is important because it relates directly to what happens to transmission following an influx of hosts. If an influx of new birds arrives and the number of mosquitoes doesn't change (see point 1) transmission could actually decrease because the number of bites per bird decreases.
4. In the description of the bird hosts in the main text and supplement, specify the infection status of all immigrating birds – is it assumed that all immigrants are uninfected?
5. Clarify the statement about 'herd immunity' in p6 of the parameterization section of the supplement: is the proportion of immune birds updated in the model each week applied only to new arrivals? I think in an earlier section you say that birds recover after 6 days. Also, because 'herd immunity' typically refers to the population-level phenomenon arising from having immune individuals rather than the % immune per se, I would rephrase to describe as 'the proportion of birds in the recovered and immune class'.
6. The model assumes that mosquito biting preferences, and mosquito mortality, are not influenced by the infection or nutritional status of the host. This could be worth specifying given empirical work demonstrating that these assumptions don't always hold, and thus could influence prevalence in mosquitoes

Minor comments

Main text

Line 169: shorten the title of this section to 'agent based model'

Line 172: add 'the' before American robin (and check for other instances of missing definite articles in the modeling and results section and supplement)

Line 240: define MIR at its first use

Lines 242-246: Modify to use a colon after the first sentence and a semicolon between the descriptions of phases 1 and 2.

Lines 306 and 314: I found the sentences about the stress levels difficult to interpret - could this be rephrased?

Supplement

page 6. I didn't understand the very last sentence on this page "With a 1-host."

Referee: 2

Comments to the Author(s).

I enjoyed reading this study by Owen et al. that examined how short-term food restriction influences West Nile virus infection dynamics for American robin reservoirs. This is a critical question because variation in food availability is arguably one of the most universal ecological stressors in natural populations, but it's a challenging stressor to study outside of experimental manipulations such as the one done here. Thus, we know very little about how variation in food access, including short-term food restriction, influence infection dynamics in wildlife. In this case, understanding infection dynamics in wildlife has important consequences for human health, given that the virus studied here is zoonotic in nature. Overall this was a very well done experiment and I particularly liked the integration of the experimental data with a mathematical model to extrapolate the within-host effects detected to disease dynamics more broadly. I have some comments that I hope will improve clarity and impact of the manuscript.

Broader comments

The authors place the study in the context of anthropogenic environmental change in several places throughout the manuscript. While I appreciate that global change will likely exacerbate variation in resource availability, what I think is particularly important about resource availability (that doesn't really come through in the manuscript, in my opinion) is that is a relatively universal stressor that can occur even in the absence of anthropogenic effects. I think the manuscript would be even stronger if that aspect came through. I would also like for the connection between food availability and migration to be made stronger in the manuscript- the authors focus on robins during the fall for good reason, but don't really build the conceptual links between their study design and when robins might experience food restriction "naturally". For example, at the end of the introduction you nicely set up your experiment and model but there is no direct connection made between robin ecology (what the likely periods of food restriction in the wild and how those map onto demography, migration) and WNV spread. Some of this comes later when you set up the model (Lines 182-185) but I think it would be much stronger to set this all up in the intro since it is really the motivation for the study design. You also mention later on that 20 Sept- 31 October is the local viral amplification phase but I don't think this comes up until Line 244-245. It would be great to have this upfront so the motivation for the timing of your experiment is understood.

I really liked that you based your model on empirical data, including banding data and serosurveillance data from nearby. I wasn't sure if some of that data needs to be analyzed or presented descriptively more formally (in the supplement), given that it motivates the model, or whether that is done elsewhere and thus not needed here?

Minor comments:

Line 16: I think "as" should be "at" in this case.

Line 27-28: duplication of "and predict"

Line 42: I wasn't sure what "top level" meant in this context- is this akin to "top down"?

Line 54-55: This is minor but this is the first use of "host resistance"- it could be clarifying to explicitly link this term back to mechanism 3 where you don't explicitly use the term resistance.

Line 57: might be helpful to add "of immune responses" after "requirements"

Line 63: this is the first use of "infectiousness"- might be helpful somewhere to clarify that here you use viral loads as a proxy for infectiousness.

Line 66: this was a bit of a rough transition for me when reading... could you instead link better to prior paragraph by starting with something like "The variation in infectiousness caused by nutrition is key to understand because variation in infectiousness at both the level of the population and individual is not well understood."

Line 77: I suggest adding "or vectors" after "susceptible hosts"

Line 81: might be helpful to clarify that here that you are looking within-species by saying "among-individual variation in..." or "within-species variation in host infectiousness" instead of just "host's infectiousness".

Line 84: I suggest deleting the word "amplified" here.

Line 90: I wouldn't have a dash between "environmental" and "determinant"

Very end of Line 140: Given that 3 birds were treated slightly differently (for good reason), I suggest adding "Aside from these three individuals,..." just before "we randomly assigned robins..."

Lines 147-151: It's clear in the figure but it would be good to make it clear in the text that birds were re-fed again starting at inoculation or just before. It is unclear in the text itself here.

Lines 200-201: I wasn't clear what happens to the robins that are both food stressed and have high viral titres- do they die? This makes sense based on your results but it isn't clear in text whether they are just not departing for migration or are dying.

Line 206: I suggest adding "that are" in front of "immune"

Line 211: I suggest adding "are" in front of "replaced"

Line 225: Technically I think this would be a mixed linear model, since a traditional ANOVA doesn't allow for random effects. But maybe this is incorrect and I just haven't heard of whatever model you used... was this done in R?

Line 252: you mention "multiple models" - is the model set listed somewhere? and how was it determined?

Line 268-269: can you clarify what you mean by "between" and "within" groups? Make clear what the "group" refers to here.

Line 272: if you give the mean in the text, I would also give some kind of metric of uncertainty like standard deviation.

Lines 277-283: there are a lot of different mass analyses- I see that they slightly differ and perhaps are all important to include but I wonder if it could be simplified in the text and some of the analyses moved to supplement so that it doesn't distract from the key analyses of how food restriction affected infection responses.

Line 290: "infectious longer" - is the unit in days? I don't think I saw how this was quantified- just the number of days over the infectious index of 4?

Line 300-315: I would add "food" in front of stress throughout this entire section just to clarify what type of "stress" you mean. Stress is such a vague term and you imposed a very specific form of it.

Lines 307-308: awkward wording of this sentence.

Line 315: also a bit awkward- perhaps different "from" each other rather than "with."

Line 322: it wasn't clear to me where the 200% came from in the results section. I like doing percentages for easier interpretation but it would help if it were set up in the results section to see what result generated this.

Line 326: I liked how you made the link to levels associated with previous WNV epidemics. Could that level be somehow added to the graph to make it visually obvious?

Line 334: I think should be "at" instead of "as":

Line 337: "biggest effect" relative to what? perhaps just say large effects to be safe.

Line 377: could you add a "were" before "puffed up"

Line 380: seems like a few words are missing (similar TO, maybe?)

Lines 383: this is an interesting link with the cort implants. Do you think cort may mediate some of the effects seen here? Do cort levels go up with food restriction? You may not want to speculate on this but the parallels were quite interesting.

Line 385: "translates" should be "translate" here

Line 391: Awkward wording- maybe say "demonstrates that increasing the proportion of the robin population experience..."

Line 402: remove either "of" or "in" at end of line

Figure comments

Figure 2 caption. Can you put the dates in for the calculation of proportional change in body mass (was that day 19 minus 14, for example)?

Figure 2. Could the treatments be organized to put the food restricted treatment next to each other, and/or with the same color?

Figure 3. I like the idea of showing the data for individual birds to see variability (Panel A) but it is hard to see which birds are in which treatment- can the points themselves be made slightly larger? Also the fact that the triangles and circles represent the treatment assignments in Panel A is not clear anywhere (needs to be in legend) and differs from the use of triangles in Panel B. To avoid confusion I would use a different type of symbol for the limit of detection and keep the triangles representing food restriction in Panel B as in Panel A.

Figure 3. Can you clarify the Y-axis label that it is per 1000 mosquitoes? Also, can Proportion stressed say Proportion "Food" Stressed for clarity?

Author's Response to Decision Letter for (RSPB-2021-0881.R0)

See Appendix B.

RSPB-2021-0881.R1 (Revision)

Review form: Reviewer 1

Recommendation

Accept with minor revision (please list in comments)

Scientific importance: Is the manuscript an original and important contribution to its field?

Excellent

General interest: Is the paper of sufficient general interest?

Excellent

Quality of the paper: Is the overall quality of the paper suitable?

Excellent

Is the length of the paper justified?

Yes

Should the paper be seen by a specialist statistical reviewer?

No

Do you have any concerns about statistical analyses in this paper? If so, please specify them explicitly in your report.

No

It is a condition of publication that authors make their supporting data, code and materials available - either as supplementary material or hosted in an external repository. Please rate, if applicable, the supporting data on the following criteria.

Is it accessible?

Yes

Is it clear?

Yes

Is it adequate?

Yes

Do you have any ethical concerns with this paper?

No

Comments to the Author

Thanks to the authors for their clarifying comments about the model structure. The responses addressed my concerns and I'm excited to see this interesting study published. I have two minor suggested edits:

Line 333: Remove the word 'fixed' (since the 10:1 ratio changes over the course o simulations with robin influxes). It might be most transparent just to state that you start the simulation with 5000 mosquitoes (representing an initial 10:1 ratio of mosquitoes to birds).

Line 340: state the units of the mosquito bite rate (e.g. 'per bird per day')

Review form: Reviewer 2

Recommendation

Accept with minor revision (please list in comments)

Scientific importance: Is the manuscript an original and important contribution to its field?

Excellent

General interest: Is the paper of sufficient general interest?

Good

Quality of the paper: Is the overall quality of the paper suitable?

Excellent

Is the length of the paper justified?

Yes

Should the paper be seen by a specialist statistical reviewer?

No

Do you have any concerns about statistical analyses in this paper? If so, please specify them explicitly in your report.

No

It is a condition of publication that authors make their supporting data, code and materials available - either as supplementary material or hosted in an external repository. Please rate, if applicable, the supporting data on the following criteria.

Is it accessible?

Yes

Is it clear?

Yes

Is it adequate?

Yes

Do you have any ethical concerns with this paper?

No

Comments to the Author

Overall I am very happy with the way that the authors revised the work- the context of the work is clearer and all of my concerns were addressed. I have some remaining editorial comments- most are very minor- but one (days post-inoculation) definitely needs to be addressed for clarity throughout the manuscript.

Comments

I like the incorporation of "days post-inoculation" but the days that you use throughout much of the paper (including in the figures) are not actually days post-inoculation, since day "0" is the day you moved the birds to New York. I would suggest making inoculation day as your day "0" for clarity (and then using -2 DPI for 2 days prior to inoculation, for example), or removing the term "days post-inoculation" and only using it when discussing viral sampling, for example (perhaps you could call the other days something else so that it's clear that those dates are prior to inoculation). This comes up in Line 385 (you state that body mass was looked at between 14 and 19 dpi but that is actually day 14 and 19 after the move, and PRIOR to inoculation). This also comes up in the captions/legends for Figure 1 and Figure 2 (which again states 14 DPI-19 DPI, which is incorrect). An easy but important fix!

Line 18: I suggest putting a comma after "approaches" for clarity.

Line 33: similarly, I suggest a comma after "significantly" for clarity.

Lines 42-44: If you need to save space further, I didn't think that this second line added much.

Line 118: there is an extra "a" before "superspreaders"

Line 134: for clarity, I suggest replacing "or" at the very end of the line with "termed" (so that it's clear that MIR is the term for what you defined rather than another thing entirely)

Line 151: "individual cages" should be deleted; right now it says "individual wire cages individual cages"

Line 154-155: this is very minor but it would be clearer if all the food info was in one place and not interrupted by water. Could you move line 155-156 ("Birds were provided ad lib access to water") up before the line "All the birds were fed a mixed diet..." since the next few sentences are also about food. Then all the food info is together.

Line 178: I think you can delete "the" before "ambient" to save words.

Line 190: would be clearer to add "one of" in front of "the two Sham groups."

Line 250: I missed this the first time, but can you clarify what you mean by "except for robins that are both food stressed and have viral titers exceeding log...". Do those robins remain in the model rather than "departing" theoretically?

Line 415: Please put commas after each set of results for clarity... aka a comma before "were infectious longer" and a comma before "and had a higher infectious index"

Line 415: when you say 200% more infectious days, can you add the means for each group and not just the stats values?

Line 480: I think a word like "Nonetheless" would be a more appropriate start than "And" to the

sentence... "while it is well understood...". It's interesting that food limitation is this universally important stressor yet we actually know very little about how it affects host defenses.

Line 483: I think the "as time of infection" should be "AT time of infection"

Line 486: period needed before "In laboratory mice"

Line 492-493: there is no close to the parenthesis ")" in the line

Line 494: the mallard sentence needs rewording to be grammatically correct- right now it reads as if the infected mallards themselves affected viral shedding. Maybe "In mallards infected with low-pathogenic avian influenza virus, chronic food restriction affected..."

Line 530: there is some kind of word missing here- maybe a "to" needed after "similar"?

Line 539: very minor but the word "model" is redundant if you say ABM so you could save a word here by deleting "model" :)

Line 543: I suggest adding a comma after "mosquitoes" for clarity.

Line 562: the word "that" needs to be deleted

Figure 3C. Is there a meaning to the "1" and "2" on the y-axis. If not, can you just instead label the bars with their treatment group and remove the figure legend? It confused me having the numbers there.

Decision letter (RSPB-2021-0881.R1)

07-Jul-2021

Dear Dr Owen

I am pleased to inform you that your manuscript RSPB-2021-0881.R1 entitled "Reservoir hosts experiencing food stress alter transmission dynamics for a zoonotic pathogen." has been accepted for publication in Proceedings B.

The referees have recommended publication, but also suggest some minor revisions to your manuscript. Therefore, I invite you to respond to the referees' comments and revise your manuscript. Because the schedule for publication is very tight, it is a condition of publication that you submit the revised version of your manuscript within 7 days. If you do not think you will be able to meet this date please let us know.

1) A text file of the manuscript (doc, txt, rtf or tex), including the references, tables (including captions) and figure captions. Please remove any tracked changes from the text before submission. PDF files are not an accepted format for the "Main Document".

2) A separate electronic file of each figure (tiff, EPS or print-quality PDF preferred). The format should be produced directly from original creation package, or original software format. PowerPoint files are not accepted.

3) Electronic supplementary material: this should be contained in a separate file and where possible, all ESM should be combined into a single file. All supplementary materials accompanying an accepted article will be treated as in their final form. They will be published alongside the paper on the journal website and posted on the online figshare repository. Files on figshare will be made available approximately one week before the accompanying article so that the supplementary material can be attributed a unique DOI.

Sincerely,

Professor Hans Heesterbeek

Reviewer(s)' Comments to Author:

Referee: 1

Comments to the Author(s)

Thanks to the authors for their clarifying comments about the model structure. The responses addressed my concerns and I'm excited to see this interesting study published. I have two minor suggested edits:

Line 333: Remove the word 'fixed' (since the 10:1 ratio changes over the course o simulations with robin influxes). It might be most transparent just to state that you start the simulation with 5000 mosquitoes (representing an initial 10:1 ratio of mosquitoes to birds).

Line 340: state the units of the mosquito bite rate (e.g. 'per bird per day')

Referee: 2

Comments to the Author(s)

Overall I am very happy with the way that the authors revised the work- the context of the work is clearer and all of my concerns were addressed. I have some remaining editorial comments- most are very minor- but one (days post-inoculation) definitely needs to be addressed for clarity throughout the manuscript.

Comments

I like the incorporation of "days post-inoculation" but the days that you use throughout much of the paper (including in the figures) are not actually days post-inoculation, since day "0" is the day you moved the birds to New York. I would suggest making inoculation day as your day "0" for clarity (and then using -2 DPI for 2 days prior to inoculation, for example), or removing the term "days post-inoculation" and only using it when discussing viral sampling, for example (perhaps you could call the other days something else so that it's clear that those dates are prior to inoculation). This comes up in Line 385 (you state that body mass was looked at between 14 and 19 dpi but that is actually day 14 and 19 after the move, and PRIOR to inoculation). This also comes up in the captions/legends for Figure 1 and Figure 2 (which again states 14 DPI-19 DPI, which is incorrect). An easy but important fix!

Line 18: I suggest putting a comma after "approaches" for clarity.

Line 33: similarly, I suggest a comma after "significantly" for clarity.

Lines 42-44: If you need to save space further, I didn't think that this second line added much.

Line 118: there is an extra "a" before "superspreaders"

Line 134: for clarity, I suggest replacing "or" at the very end of the line with "termed" (so that it's clear that MIR is the term for what you defined rather than another thing entirely)

Line 151: "individual cages" should be deleted; right now it says "individual wire cages individual cages"

Line 154-155: this is very minor but it would be clearer if all the food info was in one place and not interrupted by water. Could you move line 155-156 ("Birds were provided ad lib access to water") up before the line "All the birds were fed a mixed diet..." since the next few sentences are also about food. Then all the food into is together.

Line 178: I think you can delete "the" before "ambient" to save words.

Line 190: would be clearer to add "one of" in front of "the two Sham groups."

Line 250: I missed this the first time, but can you clarify what you mean by "except for robins that are both food stressed and have viral titers exceeding log...". Do those robins remain in the model rather than "departing" theoretically?

Line 415: Please put commas after each set of results for clarity... aka a comma before "were infectious longer" and a comma before "and had a higher infectious index"

Line 415: when you say 200% more infectious days, can you add the means for each group and not just the stats values?

Line 480: I think a word like "Nonetheless" would be a more appropriate start than "And" to the sentence... "while it is well understood...". It's interesting that food limitation is this universally important stressor yet we actually know very little about how it affects host defenses.

Line 483: I think the "as time of infection" should be "AT time of infection"

Line 486: period needed before "In laboratory mice"

Line 492-493: there is no close to the parenthesis ")" in the line

Line 494: the mallard sentence needs rewording to be grammatically correct- right now it reads as if the infected mallards themselves affected viral shedding. Maybe "In mallards infected with low-pathogenic avian influenza virus, chronic food restriction affected..."

Line 530: there is some kind of word missing here- maybe a "to" needed after "similar"?

Line 539: very minor but the word "model" is redundant if you say ABM so you could save a word here by deleting "model" :)

Line 543: I suggest adding a comma after "mosquitoes" for clarity.

Line 562: the word "that" needs to be deleted

Figure 3C. Is there a meaning to the "1" and "2" on the y-axis. If not, can you just instead label the bars with their treatment group and remove the figure legend? It confused me having the numbers there.

Author's Response to Decision Letter for (RSPB-2021-0881.R1)

See Appendix C.

Decision letter (RSPB-2021-0881.R2)

19-Jul-2021

Dear Dr Owen

I am pleased to inform you that your manuscript entitled "Reservoir hosts experiencing food stress alter transmission dynamics for a zoonotic pathogen." has been accepted for publication in Proceedings B.

Data Accessibility section

Open Access

Paper charges

Sincerely,

Proceedings B

Appendix A

Response to Editorial Board's suggestions for revision of manuscript entitled: *Reservoir hosts experiencing food stress alter transmission dynamics for a zoonotic pathogen.*

Editorial Board Comment: I think this paper would be of broad interest to readers of the journal. However, the paper currently lacks sufficient detail about the modeling approach within the main text. Most of the relevant modeling required to understand what was done appear in the supplement, and this detracts significantly from the paper. To be reviewed, the methods of the paper would need to be sufficiently described in the main text.

Author response: We agree that the modeling detail is important and should be placed in the main document. Originally when we first submitted the manuscript, it exceeded the 10-page limit and we needed to cut it by 1,700 words to submit. We did this by tightening up the whole manuscript, particularly the methods for the experiment and moving most of the modeling details into the supplemental material.

So, we agree with review board and in this revision, we have done our best to move that detail back into the paper in a condensed format that is sufficient to understand the modeling approach. We left the full and more expansive description of the modeling approach in the supplemental materials. To accommodate the additional text we then tightened/shortened the discussion and pared down references in places where there were 3 or more. We hope that our revision will address the concerns and also meet the page limit.

Appendix B

Dear Professor Heesterbeek,

We appreciate the positive and constructive review of the manuscript by the AE and the two referees. Below I respond to each point made by the Referees. The referee's comments are in *italics* and author responses are in blue font. For many of the simple comments, we just put "Done" indicating that we made the suggested change in the document. For the more substantive comments we provide more thorough responses. The major comments were about the model description and parameterization (R1) and placing the study into a broader context (R2). For the model, we were not able to provide the functional forms they were suggesting because of the nature of an agent-based model relative to a traditional equation-based model. We provide an explanation below about why there are no functional forms for the parameters mentioned by R1. However, R1 did point out some mistakes (particularly regarding mosquito population size) made in our description which we corrected accordingly.

For the additional context suggested by R2, which were fantastic points – we did what we could to modify/amend the text to highlight or capture those points. Our hope is that the additional text will not put us over the page limit. If it does, we will need to adjust as needed. I would hesitate to remove much of the current text given how much it was pared down from original submission due to page limits. However, R2 also recommended removing some results and placing into the supplemental file, which saved us space. With the edits to the manuscript and formatting the word count went from 7771 words to 8492 (7950 without formats to bibliography).

062321: The manuscript was returned for being too long. We had to pare down some of our initial changes/additions and worked on correcting bibliography. It is now 7743 words. The places we pared down text was in the additional text about migration in the introduction and the description of the model. Despite these changes we feel we were still able to address the referee comments/suggestions.

Referee: 1

I found this paper to be interesting and exciting, and worthy of publication in Proceedings B. My comments below are focused on the reporting of the agent-based model in the main text and results. Overall this is well described, but I have a few suggestions and clarifying comments below, with the most important point to address relating to assumptions about the transmission rate and how mosquito abundance changes with bird abundance.

Please note that while I addressed the Referee 1's comments individually, there is a lot of overlap in the comments and my responses. It is my hope that the responses in their entirety, adequately address the Referee's concerns/suggestions.

Clarify assumptions about mosquito abundance. In the supplement (c-mortality) you say that the abundance of mosquitoes is fixed over time, but in the main text and elsewhere it says that the relative abundance of mosquitoes to birds is fixed at a 10:1 ratio. So, what happens to mosquito abundance during the large influx of robins on day 50 and day 71? If you assume that mosquito

abundance also jumps with the influx of birds on these days you should make this assumption explicit, and justify why it's reasonable to assume that mosquito abundance tracks bird abundance through time in this way. You should briefly discuss the potential consequences of this assumption for transmission in the discussion (see point 3 on transmission rate below)

Response: Our statement "We incorporated a density dependent mortality rate of 0.04 (52) to limit the growth of the mosquito population" was inaccurate and we have corrected this mistake in the main document. Density-dependence is not incorporated in the model. The culex population size is determined during model setup using the user-provided culex-to-host-ratio, in this case it was set at 10:1. The culex abundance remains constant throughout the model run and does not change with the influx of robins during the season; hence, in this study the mosquito population is fixed at 5,000 regardless of when robin population increases above 500. In the model run we essentially simulate Culex population turnover throughout the model run with all culex that die being replaced by an adult culex. One reason we chose not to let mosquito population fluctuate with robin numbers was because of the natural (temp/season) decline in mosquito population from summer to fall. Had we maintained a constant 10:1 mosquito: host ratio we would have over-inflated the true entomological risk.

2. For replicability, it's important to show the functional forms used to describe any transition probabilities relating to demography and infection that are not constant per capita rates in the supplement (e.g. how exactly are the effects of density-dependent mosquito mortality and temperature-dependence in transmission modelled?).

Response: We appreciate this comment however, with this being an agent-based model (ABM) and a simple one, there is no functional form of transmission - the ABMs use a bottom-up approach. Agents and their interactions are guided by a set of inductively generated local rules and behaviors and these in turn give rise to emergent phenomena at a group or system-wide level. So functional form of transmission is not enforced in this model, but in fact, is an emergent property of the model. We used very simple rules for this model and the parameter values we used for these "rules" were derived from the literature and not individually modeled in this simulation. The simple rules we have used are supported by published literature (see the **Parameterization and calibration** section and the associated references). Further, the model is currently published (Aniruddha Belsare, Jennifer Owen (2021)) and one can look at the code of the model for comments to help the user understand these processes. Regardless, we did edit the supplemental file to help guide the reader.

The format of the model explanation is using the ODD protocol which varies from how other epidemiological models would be described. See Grimm et al. 2020 - <http://jass.soc.surrey.ac.uk/23/2/7.html> for explanation of the ODD protocol for describing ABMs.

Crucially, the functional form of the transmission rate between host and vectors needs to be specified. Did you use the standard Ross-McDonald transmission function for vector-borne

transmission, which assumes that bites are distributed among hosts and thus the number of bites per host decreases with increasing bird density? Or alternatively did you use a simple mass-action term (and if so, justify why)? This specification is important because it relates directly to what happens to transmission following an influx of hosts. If an influx of new birds arrives and the number of mosquitoes doesn't change (see point 1) transmission could actually decrease because the number of bites per bird decreases.

Response: See answer above about mortality. There is no density-dependent mosquito mortality in the model. The temperature dependence in transmission is not modeled explicitly, but we have parameterized the model using temperature-specific biting rate (see Parameterization and calibration section of supplement) based on the literature for 23°C. Additionally, we included the length of time for the mosquito to be capable of transmitting virus (duration to fully disseminate) called the extrinsic incubation period which ranges from 7 to 11 days, values also derived from the literature and associated with temperature of 23°C.

We understand the reviewer's comment about dilution effect with the lower Culex to robin ratio with influx of birds but this assumption is not necessarily true when dealing with an ABM. In an ABM an infected mosquito can bite many susceptible or infected birds and with more robins, it may (or may not) decrease the chance of a mosquito taking a blood meal from the same bird.

In the description of the bird hosts in the main text and supplement, specify the infection status of all immigrating birds – is it assumed that all immigrants are uninfected?

Response: Yes, they are all uninfected, clarified this in the main text and the supplement.

Clarify the statement about 'herd immunity' in p6 of the parameterization section of the supplement: is the proportion of immune birds updated in the model each week applied only to new arrivals? I think in an earlier section you say that birds recover after 6 days. Also, because 'herd immunity' typically refers to the population-level phenomenon arising from having immune individuals rather than the % immune per se, I would rephrase to describe as 'the proportion of birds in the recovered and immune class'.

Response: We agree the use of 'herd immunity' was incorrectly used and have clarified the language to indicate proportion of birds that are immune. Yes, the immune status is based on new arrivals given there is a complete turnover of robins (except ones that meet criteria described in the model parametrization).

The model assumes that mosquito biting preferences, and mosquito mortality, are not influenced by the infection or nutritional status of the host. This could be worth specifying given empirical work demonstrating that these assumptions don't always hold, and thus could influence prevalence in mosquitoes

Response: Good point and we added some text in the main document to indicate that this assumption may not hold in nature.

Referee 1: Minor comments

Main text

Line 169: shorten the title of this section to 'agent based model'

Done

Line 172: add 'the' before American robin (and check for other instances of missing definite articles in the modeling and results section and supplement)

Done

Line 240: define MIR at its first use

Done

Lines 242-246: Modify to use a colon after the first sentence and a semicolon between the descriptions of phases 1 and 2.

Done: Line 317

Lines 306 and 314: I found the sentences about the stress levels difficult to interpret – could this be rephrased?

Done: Lines 346 – 348.

Supplement

page 6. I didn't understand the very last sentence on this page "With a 1-host."

Done: revised this sentence

Referee: 2

I enjoyed reading this study by Owen et al. that examined how short-term food restriction influences West Nile virus infection dynamics for American robin reservoirs. This is a critical question because variation in food availability is arguably one of the most universal ecological stressors in natural populations, but it's a challenging stressor to study outside of experimental manipulations such as the one done here. Thus, we know very little about how variation in food access, including short-term food restriction, influence infection dynamics in wildlife. In this case, understanding infection dynamics in wildlife has important consequences for human health, given that the virus studied here is zoonotic in nature. Overall this was a very well done experiment and I particularly liked the integration of the experimental data with a mathematical model to extrapolate the within-host effects detected to disease dynamics more broadly. I have some comments that I hope will improve clarity and impact of the manuscript.

The authors place the study in the context of anthropogenic environmental change in several places throughout the manuscript. While I appreciate that global change will likely exacerbate variation in resource availability, what I think is particularly important about resource availability (that doesn't really come through in the manuscript, in my opinion) is that is a relatively universal stressor that can occur even in the absence of anthropogenic effects. I think the manuscript would be even stronger if that aspect came through.

Response: Great point and we did add some text in the abstract, intro, and discussion to indicate it is a stressor that is not wholly a consequence of human activities. I had to be very judicious with additional text given the strict page limits and due to the next great suggestion by the referee, we saved our space so we could accommodate the additional content needed to address the linkages to migration.

I would also like for the connection between food availability and migration to be made stronger in the manuscript- the authors focus on robins during the fall for good reason, but don't really build the conceptual links between their study design and when robins might experience food restriction "naturally". For example, at the end of the introduction you nicely set up your experiment and model but there is no direct connection made between robin ecology (what the likely periods of food restriction in the wild and how those map onto demography, migration) and WNV spread. Some of this comes later when you set up the model (Lines 182-185) but I think it would be much stronger to set this all up in the intro since it is really the motivation for the study design.

Response: Such a great comment and highlights why the review process is so important – this is something that should be in the paper and got overlooked in the revision process and by the authors being too close to the manuscript that we could not see this omission. To maintain the length of the paper the additional text is only a few sentences but hopefully highlights the relevance of the study to the migratory period, when availability food resources may be unpredictable and varied. I placed most of the additional text in the introduction and only added a few words in the discussion.

** There is a concern that this will get booted back to me because the additional text did increase total words from 7800 to 7972.

062321: we had to revise/pare down the text we added, which was as follows: *Specifically, we developed an ABM to simulate enzootic transmission of WNV in an avian host (American robin) - mosquito vector (Culex spp.) system during the fall when migratory populations of robins are migrating south to their non-breeding grounds. The energetic cost of migration is high and migratory landbirds, including robins, rely on finding suitable stopover sites where they can rest and replenish depleted fat stores [37]. Yet, for myriad reasons, such as habitat loss and degradation and climate change, the availability of food can be unpredictable and/or insufficient [38, 39]. This food limitation not only has fitness consequences for the migrant [37], it also has implications for local transmission dynamics, which we explore further with the ABM. Using a combination of empirical and experimental infection data, and evaluated alternate scenarios to test the effect of food stress on the population prevalence of WNV-infected Culex mosquitoes, or mosquito infection rate, which is a predictor of human risk [40].*

You also mention later on that 20 Sept- 31 October is the local viral amplification phase but I don't think this comes up until Line 244-245. It would be great to have this upfront so the motivation for the timing of your experiment is understood.

Response: If I understand the comment, our use of 'local viral amplification' was not well explained in the model data analysis section. We are referencing what is occurring in the simulation. In nature, this amplification would be starting earlier in the season when the virus is naturally cycling between the host and vector. Because we did not start the simulation with infected mosquitoes that period after we initiate the model run would differ from later in the season and that difference may overshadow any differences relevant to the hypothesis we are testing. I reworded that slightly by adding "in the simulation" so we are not suggesting this is the time of viral amplification in nature.

I really liked that you based your model on empirical data, including banding data and serosurveillance data from nearby. I wasn't sure if some of that data needs to be analyzed or presented descriptively more formally (in the supplement), given that it motivates the model, or whether that is done elsewhere and thus not needed here?

Response: We have provided the capture and age data for American robins at the banding station referenced in the manuscript. This data has been uploaded to Dryad. The seropositivity comes from multiple sources, including this manuscript (for HY birds). The remaining information is from sampling efforts by Dupuis (Griffen Lab) over many years and is not consolidated into one dataset.

Minor comments:

Line 16: I think "as" should be "at" in this case.

Line 27-28: duplication of "and predict"

Line 42: I wasn't sure what "top level" meant in this context- is this akin to "top down"?

Done – Line 42

Line 56-57: This is minor but this is the first use of "host resistance"- it could be clarifying to explicitly link this term back to mechanism 3 where you don't explicitly use the term resistance.

Response: I edited this slightly to link it to previous statement

Line 57: might be helpful to add "of immune responses" after "requirements"

Done

Line 63: this is the first use of "infectiousness"- might be helpful somewhere to clarify that here you use viral loads as a proxy for infectiousness.

Response: I decided to take out the term infectiousness here and edited the sentence to be clearer. See lines 68-69

Line 66: this was a bit of a rough transition for me when reading... could you instead link better to prior paragraph by starting with something like "The variation in infectiousness caused by nutrition is key to understand because variation in infectiousness at both the level of the population and individual is not well understood."

Response: not sure this reads well for me – The nutritional basis...?

Line 77: I suggest adding "or vectors" after "susceptible hosts"

Done

Line 81: might be helpful to clarify that here that you are looking within-species by saying "among-individual variation in..." or "within-species variation in host infectiousness" instead of just "host's infectiousness".

Done

Line 84: I suggest deleting the word "amplified" here.

Done

Line 90: I wouldn't have a dash between "environmental" and "determinant"

Done

Very end of Line 140: Given that 3 birds were treated slightly differently (for good reason), I suggest adding "Aside from these three individuals,..." just before "we randomly assigned robins..."

Done

Lines 147-151: It's clear in the figure but it would be good to make it clear in the text that birds were re-fed again starting at inoculation or just before. It is unclear in the text itself here.

Done

Lines 200-201: I wasn't clear what happens to the robins that are both food stressed and have high viral titres- do they die? This makes sense based on your results but it isn't clear in text whether they are just not departing for migration or are dying.

Response: I did not edit this too much since the subsequent sentence starts with "If they survive...". Then later in paragraph it mentions the mortality rules for the simulation ".. 10% of the stressed robins with infectious viremia levels die on the 5th day of infection". To clarify, I added (see below for mortality of infectious birds) to highlight that not all the stressed infectious birds die.

Line 206: I suggest adding "that are" in front of "immune"

Response: I did not see where this needed to be changed and if it is in the sentence I think they are referencing (my numbers do not line up) then the sentence would not read like it is intended with this edit.

Line 211: I suggest adding "are" in front of "replaced"

Response: My version has "are" in it - unless I am looking at wrong line?

Line 225: Technically I think this would be a mixed linear model, since a traditional ANOVA doesn't allow for random effects. But maybe this is incorrect and I just haven't heard of whatever model you used... was this done in R?

Response: For the mass and viremia analysis we did use what I have known as a mixed model ANOVA, that is a combination of between – group ANOVA and a within-group ANOVA. It was done using Systat and not R.

Line 252: you mention "multiple models" - is the model set listed somewhere? and how was it determined?

Done: We primarily revised this in the supplemental material and then summarized that we used AIC selection criterion to select the best model. We have also provided all the R code for the analysis in Dryad (see data accessibility statement).

Line 268-269: can you clarify what you mean by "between" and "within" groups? Make clear what the "group" refers to here.

Done: I clarified this on my line 339 -340. "There was a significant main effect between groups (FR and NORM) and across time within groups (both $P < 0.001$) with a statistically significant interaction ($F_{1, 80} = 13.81, P < 0.001$; Figure 3)."

Line 272: if you give the mean in the text, I would also give some kind of metric of uncertainty like standard deviation.

Done

Lines 277-283: there are a lot of different mass analyses- I see that they slightly differ and perhaps are all important to include but I wonder if it could be simplified in the text and some of the analyses moved to supplement so that it doesn't distract from the key analyses of how food restriction affected infection responses.

Done: After the key results of mass changes relative to food restriction, I summarized the additional analyses in one sentence and moved the details to the supplemental file. Making this move helped with maintaining page limit given the additional content recommended by the referee.

Line 290: "infectious longer" - is the unit in days? I don't think I saw how this was quantified- just the number of days over the infectious index of 4?

Done: I clarified that it was "in days" (line 362)

Line 300-315: I would add "food" in front of stress throughout this entire section just to clarify what type of "stress" you mean. Stress is such a vague term and you imposed a very specific form of it.

Done

Lines 307-308: awkward wording of this sentence.

Done: I agree with referee that it was awkward and revised the sentence universal

Line 315: also a bit awkward- perhaps different "from" each other rather than "with."

Done

Line 322: it wasn't clear to me where the 200% came from in the results section. I like doing percentages for easier interpretation but it would help if it were set up in the results section to see what result generated this.

Done: added the 200% to the relevant section in the results.

Line 326: I liked how you made the link to levels associated with previous WNV epidemics. Could that level be somehow added to the graph to make it visually obvious?

Response: I understand why this would be an interesting addition but given the levels associated with epidemics vary with geographic location, avian community, vector species, it would be difficult to adequately capture that in the figure. Our intent was to demonstrate that our MIR values have epidemiological significance.

Line 334: I think should be "at" instead of "as:"

Line 337: "biggest effect" relative to what? perhaps just say large effects to be safe.

Done

Line 377: could you add a "were" before "puffed up"

Done

Line 380: seems like a few words are missing (similar TO, maybe?)

Done

Lines 383: this is an interesting link with the cort implants. Do you think cort may mediate some of the effects seen here? Do cort levels go up with food restriction? You may not want to speculate on this but the parallels were quite interesting.

Response: While we have not looked at Cort relative to food restriction specifically, we have seen another proxy of stress, heterophil to lymphocyte ratio be higher in birds in poor body condition (low fat stores). We agree that it is interesting and warrants further investigation; however, it is difficult to tease these apart in experimental infections due to the sampling challenges.

Line 385: "translates" should be "translate" here

Done

Line 391: Awkward wording- maybe say "demonstrates that increasing the proportion of the robin population experience..."

Done – we modified the suggested revision slightly to be “...demonstrates that increasing the proportion of the robin population experiencing “food stress”...”

Line 402: remove either "of" or "in" at end of line

Done

Figure comments

Figure 2 caption. Can you put the dates in for the calculation of proportional change in body mass (was that day 19 minus 14, for example)?

Done

Figure 2. Could the treatments be organized to put the food restricted treatment next to each other, and/or with the same color?

Done: Great suggestion by the referee and along with the following comment, I edited all the figures to keep the symbols consistent between the figures.

Figure 3. I like the idea of showing the data for individual birds to see variability (Panel A) but it is hard to see which birds are in which treatment- can the points themselves be made slightly larger?

Done

Also the fact that the triangles and circles represent the treatment assignments in Panel A is not clear anywhere (needs to be in legend) and differs from the use of triangles in Panel B.

Done: added info into legend and revised figures to be consistent with symbols.

To avoid confusion I would use a different type of symbol for the limit of detection and keep the triangles representing food restriction in Panel B as in Panel A.

Done – changed to squares.

Figure 3. Can you clarify the Y-axis label that it is per 1000 mosquitoes? Also, can Proportion stressed say Proportion "Food" Stressed for clarity?

Done

Appendix C

Dear Professor Heesterbeek,

We appreciate the positive and constructive review of the manuscript by the AE and the two referees. Below I respond to each point made by the Referees. The referee's comments are in *italics* and author responses are in blue font. For many of the simple comments, we just put "Done" indicating that we made the suggested change in the document.

When I was addressing R2's comments about the Day 14 and Day 19 vs. reporting in dpi, I noticed something that none of us caught in previous versions. Figure 2 did not match how we presented the results in the main text. In the results section we reported the change in body mass using the two-way repeated measures ANOVA but the figure was depicting results of a two-way ANOVA (non-repeated measures). So, I replaced the graph with one that matched the main body text. I went back to review the previous reviews by referees of the old Figure 2 and made sure I maintained suggested edits – i.e., consistency in symbols and colors between figures and organization of data on graph (keep food restricted groups on one side of graph and normal fed on other side of graph), and use of dpi vs. Day.

Referee: 1

Line 333: Remove the word 'fixed' (since the 10:1 ratio changes over the course o simulations with robin influxes). It might be most transparent just to state that you start the simulation with 5000 mosquitoes (representing an initial 10:1 ratio of mosquitoes to birds).

Done

Line 340: state the units of the mosquito bite rate (e.g. 'per bird per day')

Done

Referee: 2

Comments

I like the incorporation of "days post-inoculation" but the days that you use throughout much of the paper (including in the figures) are not actually days post-inoculation, since day "0" is the day you moved the birds to New York. I would suggest making inoculation day as your day "0" for clarity (and then using -2 DPI for 2 days prior to inoculation, for example), or removing the term "days post-inoculation" and only using it when discussing viral sampling, for example (perhaps you could call the other days something else so that it's clear that those dates are prior to inoculation). This comes up in Line 385 (you state that body mass was looked at between 14 and 19 dpi but that is actually day 14 and 19 after the move, and PRIOR to inoculation). This also comes up in the captions/legends for Figure 1 and Figure 2 (which again states 14 DPI-19 DPI, which is incorrect). An easy but important fix!

The referee picked up an issue in how days and dpi were being used. After taking multiple approaches, I ended up changing everything to dpi and removing "Day" altogether. This meant revising Figure 1 timeline, Figure 2 legend and axes, and changing throughout the paper.

Line 18: I suggest putting a comma after "approaches" for clarity.

Done

Line 33: similarly, I suggest a comma after "significantly" for clarity.

Done

Lines 42-44: If you need to save space further, I didn't think that this second line added much.

Done

Line 118: there is an extra "a" before "superspreaders"

Done

Line 134 (my 110): for clarity, I suggest replacing "or" at the very end of the line with "termed" (so that it's clear that MIR is the term for what you defined rather than another thing entirely)

Done

Line 151: "individual cages" should be deleted; right now it says "individual wire cages individual cages"

Done

Line 154-155: this is very minor but it would be clearer if all the food info was in one place and not interrupted by water. Could you move line 155-156 ("Birds were provided ad lib access to water") up before the line "All the birds were fed a mixed diet..." since the next few sentences are also about food. Then all the food info is together.

Done

Line 178 (142): I think you can delete "the" before "ambient" to save words.

Done

Line 190 (154): would be clearer to add "one of" in front of "the two Sham groups."

Done

Line 250 (203 – 205): I missed this the first time, but can you clarify what you mean by "except for robins that are both food stressed and have viral titers exceeding log...". Do those robins remain in the model rather than "departing" theoretically?

This was confusing for both referees so I revised the sentence for clarity -See lines 205 – 207.

Line 415: Please put commas after each set of results for clarity... aka a comma before "were infectious longer" and a comma before "and had a higher infectious index"

Done

Line 415 (292): when you say 200% more infectious days, can you add the means for each group and not just the stats values?

I was not exactly sure what they meant but I did add the actual number of infectious days for each group. "...200% more infectious days, 24 days for FR_WNV and 12 days for Norm_WN; ..."

Line 480 (334): I think a word like "Nonetheless" would be a more appropriate start than "And" to the sentence... "while it is well understood...". It's interesting that food limitation is this universally important stressor yet we actually know very little about how it affects host defenses.

Done

Line 483: I think the "as time of infection" should be "AT time of infection"

Done

Line 486: period needed before "In laboratory mice"

Done

Line 492-493 (348): there is no close to the parenthesis ")" in the line

Done

Line 494: the mallard sentence needs rewording to be grammatically correct- right now it reads as if the infected mallards themselves affected viral shedding. Maybe "In mallards infected with low-pathogenic avian influenza virus, chronic food restriction affected..."

Done

Line 530: there is some kind of word missing here- maybe a "to" needed after "similar"?

Done

Line 539: very minor but the word "model" is redundant if you say ABM so you could save a word here by deleting "model" :)

Done

Line 543: I suggest adding a comma after "mosquitoes" for clarity.

Done

Line 562: the word "that" needs to be deleted

Done

Figure 3C. Is there a meaning to the "1" and "2" on the y-axis. If not, can you just instead label the bars with their treatment group and remove the figure legend? It confused me having the numbers there.

Done